Synchronised provisioning at the nest: parental coordination over care in a socially monogamous species

van Rooij Erica P.
Griffith Simon C. simon.griffith@mq.edu.au
Department of Biological Sciences, Macquarie University , Sydney , Australia
Garant Dany
Electronic publication date: 2013 Dec 19
Publication date: 2013
Volume: 1
Electronic Location ID: e232
Received 2013 Oct 30; Accepted 2013 Dec 4
Copyright: © 2013 van Rooij and Griffith
Copyright year: 2013
Copyright holder: van Rooij and Griffith
License: This is an open access article distributed under the terms of the Creative Commons Attribution License, which permits unrestricted use, distribution, and reproduction in any medium, provided the original author and source are credited.
License URL: https://creativecommons.org/licenses/by/3.0/

Keywords: Biparental care, Nestling provisioning, Long-tailed finch, Poephila acuticauda, Cooperative behaviour

Funding: Australian Research Council grant #DP0881019 This study was supported by an ARC-Discovery Project grant awarded to SCG and a Macquarie University Research Excellence Scholarship to EPvR. The funders had no role in study design, data collection and analysis, decision to publish, or preparation of the manuscript.

==============================
Bi-parental care is very common in birds, occurring in over 90% of species, and is expected to evolve whenever the benefits of enhanced offspring survival exceed the costs to both parents of providing care. In altricial species, where the nestlings are entirely dependent on the parents for providing food until fledging, reproductive success is related to the capacity of the parents to provision the offspring at the nest. The degree to which parents synchronise their visits to the nest is rarely considered by studies of bi-parental care, and yet may be an important component of parental care, affecting the outcome of the reproductive attempt, and the dynamics of sexual conflict between the parents. Here we studied this aspect of parental care in the long-tailed finch (Poephila acuticauda), a socially monogamous estrildid finch. We monitored parental nest visit rates and the degree of parental visit synchrony, and assessed their effects on reproductive success (e.g., brood size, number of offspring fledged and nestling growth).

The frequency of nest visits in a day was low in this species (<1 visit/h), but there was a high level of synchrony by the two partners with 73% of visits made together. There was a correlation between the proportion of visits that were made by the pair together and the size of the brood at hatching, although it was not related to the number of fledglings a pair produced, or the quality of those offspring. We suggest that nest visit synchrony may primarily be driven by the benefit of parents being together whilst foraging away from the nest, or may reduce nest predation by reducing the level of activity around the nest throughout the day.

Introduction

Parental care is common in birds, with bi-parental care occurring in more than 90% of species, and expected to evolve whenever the benefits of enhanced offspring survival exceed the costs to both parents of providing care (Clutton-Brock, 1991; Royle, Smiseth & Kölliker, 2012). In altricial species, where the nestlings are entirely dependent on the parents for food until fledging, reproductive success is often limited by parental feeding rates (Royle, Hartley & Parker, 2006). However, provisioning involves energy expenditure by the parents, which may have a negative effect on their future reproduction through trade-offs with survival (Nur, 1988) or attractiveness (Griffith, 2000). Since each partners’ own future potential might be enhanced if the other parent contributed more of the total investment in offspring, there is an interesting conflict between the sexes (Trivers, 1972; McNamara et al., 2003).

Studies of socially monogamous birds have provided good opportunities to explore the evolutionary dynamics at the heart of this social bond between male and female partners (Royle, Hartley & Parker, 2002). Over the past couple of decades, possibly because of the interest in sexual conflict suggested by Triver’s (1972) classic paper, much of the research into bi-parental care has focused on the sources of variation in the level of care provided by individual males and females, particularly in the context of theoretical ideas such as the good-parent hypothesis (Hoelzer, 1989), and the differential allocation hypothesis (Burley, 1988). These hypotheses and much of the work that has followed (Royle, Hartley & Parker, 2002) has focused on the different investment strategies of males and females and the conflict between the sexes. Across all socially monogamous avian species, there is great variation in the level of relative investment by males and females with great inequity in nestling provisioning rates by the two sexes in some species (Sanz & Tinbergen, 1999; Bulit, Palmerio & Massoni, 2008), while in others investment is more equitable (Tremont & Ford, 2000; Royle, Hartley & Parker, 2006; Lee, Kim & Hatchwell, 2010).

In contrast to the focus on sexual conflict, rather less work has focused on the strategies through which parents enhance cooperation and coordination of their common goal (the production of offspring in the short- or long-term). Although widely neglected by those studying socially monogamous species with biparental care, those studying cooperatively breeding birds have understandably devoted more effort to understanding the more cooperative elements of parental care. For example, in several cooperatively breeding avian species synchronized feeding visits by helpers-at-the-nest have been observed and discussed (Doutrelant & Covas, 2007; McDonald et al., 2008; Raihani et al., 2010; Nomano et al., 2013). A number of adaptive explanations for this coordination have been proposed, such as a reduction in activity around the nest to reduce exposure to predators (Raihani et al., 2010); signaling of investment to other group members (Doutrelant & Covas, 2007; McDonald et al., 2008; Nomano et al., 2013); to enhance the distribution of food amongst offspring (Shen et al., 2010); or improve information amongst parents (Johnstone & Hinde, 2006). Alternatively, it may be that the coordination of chick feeding visits to the nest is just a by-product of normal social foraging or movement behavior to increase efficiency and reduce predation risk, as is often seen in such social species (Lee, Kim & Hatchwell, 2010; Sorato et al., 2012).

Although the social coordination of parental care in bi-parental socially monogamous species has been widely neglected, a few relatively recent papers indicate the potential importance of this aspect of parental investment. The benefits of collaborative tactics between partners have been identified in nest site selection (Stamps et al., 2002); through the speed at which egg laying is initiated (Adkins-Regan & Tomaszycki, 2007); and overall reproductive success (van de Pol et al., 2006; Mariette & Griffith, 2012). Several of these studies have indirectly looked at pair coordination by assuming that partners that have prior experience with each other are likely to be more coordinated. In addition, a couple of studies have shown directly that the behavioural compatibility of a male and female has an important effect on reproductive success (Spoon, Millam & Owings, 2006; Schuett, Dall & Royle, 2011), underlying the importance of considering the pair as a whole in addition to just the characteristics of the individuals involved.

Whilst good parental care can be achieved by two individuals working more or less independently (but contributing to the common goal of investing in the brood), recent work has found that in some species with bi-parental care, the male and female are very coordinated in their behavior. For example in a recent study of the zebra finch (Taeniopygia guttata), male and female typically visited the nest relatively few times throughout the day but with a high degree of synchrony, both visiting the nest together (Mariette & Griffith, 2012). Furthermore, during incubation in the zebra finch, both the male and female frequently acted as a sentinel for their partner while he/she was in the nest, warning of approaching danger (Mainwaring & Griffith, 2013). Such cooperative and coordinated aspects of parental care have been very rarely reported in socially monogamous species with bi-parental care. The fact that such things have only very recently been reported in the zebra finch, which is one of the intensively studied species with respect to parental care (Griffith & Buchanan, 2010), perhaps illustrates the extent to which this in a neglected area of research. It is now important to investigate the nature of the social bond in additional species with bi-parental care and consider the cooperative aspects as well as the areas of evolutionary conflict to redress the bias that has existed over the past few decades (Roughgarden, 2012).

Here we present one of the first detailed investigations of the coordination of parental care in a socially monogamous passerine with biparental care – the long-tailed finch (Poephila acuticauda). The long-tailed finch is an endemic Australian estrildid finch that is ecologically similar to the zebra finch although it inhabits the tropical savannah in the north of Australia, rather than the more arid open country that is home to the zebra finch (Higgins, Peter & Cowling, 2006). Long-tailed finches are primarily granivorous, but supplement their diet with small invertebrates (Higgins, Peter & Cowling, 2006), particularly during breeding. They are socially monogamous and pair bonds in this species are strong and durable (Zann, 1977; van Rooij & Griffith, 2011). They preferentially nest in cavities and breed readily in artificial nest-boxes (van Rooij & Griffith, 2011). Both the male and female contribute equally to nest building and the incubation of the eggs (van Rooij & Griffith, 2011), but to date there have been no detailed descriptions of offspring provisioning behaviour. Therefore, here we describe parental nest visit rates and visit synchrony in this species and examine potential effects on breeding success, and the development and condition of nestlings. We also investigated the extent to which visit rate and synchrony were predicted by social factors such as breeding density and the duration of the pair bond. Breeding density might influence the alternative opportunities that individuals have to forage socially with other adults breeding nearby (Lee, Kim & Hatchwell, 2010). The duration of the pair bond will reflect the level of previous breeding experience with the current partner (van de Pol et al., 2006) and this may affect their degree of behavioural coordination.

Methods

Study area and species

During the breeding season of 2009 (early March till late September), data was collected on long-tailed finches breeding near Wyndham, in northwest Australia (S15°33′38″, E128°08′59″). All of the pairs in this study nested in wooden nestboxes that were erected to facilitate the study of both Gouldian finch (Erythrura gouldiae) and long-tailed finches in this area (see Brazill-Boast et al., 2011; van Rooij & Griffith, 2011). Adults were caught with hand nets on their nests or using mistnets at creeks and water holes near nesting sites. All adult birds were banded with an individually numbered metal band (supplied by the Australian Bird and Bat Banding Scheme) and individual colour combinations, and a blood sample was taken from the brachial vein. Adult long-tailed finches are only slightly dimorphic but all birds studied here had been sexed using a molecular marker as part of an earlier study (van Rooij & Griffith, 2010). Pairs can raise multiple broods per season, with brood size varying from two to seven chicks (4.3 ± 1.0) being provisioned in the nest for a period of about 20.6 ± 2 days (van Rooij & Griffith, 2011). Fifty five banded pairs were studied in total with 28 pairs (51%) making only one recorded breeding attempt, 14 pairs (25%) having two, and 13 pairs making three breeding attempts (24%). We counted eggs and inspected nests daily at the predicted end of incubation period to count the number of hatchings. Nestlings were individually marked two days after hatching by clipping the end of one of the claws on their toes and subsequently monitored until they fledged. At the age of ten days all nestlings were banded, measured and weighed and a small blood sample was taken from the brachial vein and stored in ethanol. Nestlings were also measured (mass and tarsus) on day 16 (just before fledging). Nestling measures of day 16 were used as a measure of offspring quality, and we assessed the size difference (in mass) between nestlings as a percentage difference between smallest and largest offspring, to determine the extent to which parents produced a brood of even quality, on the assumption that uneven brood quality and partial brood mortality is a sign of poor parental care. We also computed residual mass to account for variation in skeletal size amongst chicks and calculated the difference in condition between the best and the poorest nestling within a brood. Local breeding density, for each reproductive attempt, was calculated as the number of active nests in the same area (see below) as the focal nest, over the days on which the young were being provisioned in the active nest. The study was conducted in a nest box breeding population with colonies of breeding birds in patches of suitable habitat that were spatially separated by areas of more open savannah with few mature trees large enough to erect nest boxes or contain natural nest cavities (further details on the study area are given in Brazill-Boast et al., 2011).

Pair duration was categorised in three ways, and based on our intensive study of all breeding activity in this area starting in January 2008 (Brazill-Boast et al., 2011; van Rooij & Griffith, 2011): (a) whether the pair bred together before (0-never recorded breeding together before; 1 recorded breeding together at least once before), (b) the number of seasons a pair had been recorded breeding together (0, 1, 2; including the current season if the pair bred together earlier in the current season) and (c) the number of times the pair had been recorded breeding together over the seasons (range 0–4). Both breeding site fidelity and partner fidelity are high in this species (van Rooij & Griffith, 2011). Ages were based on the number of years since an individual was first banded as a nestling or adult and therefore were ‘minimum ages’ and for breeding adults varied between 1 and 3 years.

Parental nest visit rates

To assess the rate of parental feeding visits we used video cameras (AVC 647 Color IR Camera; 1–2 m from the entrance of the nest box and connected to a hard-drive Archos 605 WIFI), which filmed the entrance of the nestbox. To allow easy individual recognition of the parents when entering the nest, one of the parents was marked with a white dot on the back of the head (correction fluid), two to six days before the parents were filmed (when they were captured when the chicks were about 6 to 9 days old). Although these marks did wear off within a few weeks, they were usually quite apparent on the films, and we only used data from pairs where the sexes could be readily distinguished in the videos. Birds were acclimated to the presence of the camera over a minimum 24-h period prior to recording. A total of 37 nesting attempts were filmed from 29 independent pairs (29 first breeding attempts; six pairs also on their second attempt and two pairs on their third attempt). The data collected on second and third broods were used in only one analysis comparing later broods to the first brood.

Nests were filmed when nestlings were ten and eleven days old, coinciding with the period of maximal nestling growth. Recording started around 6 am (just after sunrise) and continued for around 10 h per day (total 367 h filmed; 594 min ± 74 min per nest per day). The videos were later analysed using VLC media player, which provided accurate data on timings of behaviour in the video files. The number of visits to the nest by each parent was recorded, along with the timings of each visit and the time spent inside the nest. For rate, the number of visits was divided by the number of hours a particular nest was videoed. Whilst it was not captured in video data (which was focused on a small area around the nest), in the course of our nest monitoring and adult trapping we witnessed over a hundred incidences of adults arriving to the nest tree. Whilst this was not quantified, we also present some anecdotal observations of this behaviour.

Nest visit synchrony

We considered that males and females were together at the nest (i.e., a synchronous visit) if the second individual to enter the nest did so within 5 min of the first individual entering the nest. Given the very long intervals between visits (see results) we believe that this is justified and suggests that they have arrived at the nest tree in a coordinated way. Using the duration of nest visits and their frequency throughout the day we calculated the probability of temporally coincident visits by two parents working completely independently.

We calculated the proportion of synchronised visits as: (number of visits together × 2)/total number of visits by individuals alone. We calculated the likelihood of both parents being at the box together by chance on the basis of the actual time that each individual was present during the day. We calculated expected time spent at the nest together for each pair separately and compared them with observed proportion of time spent at the nest together with paired t-tests.

Results

There was a positive correlation between the visit rate of the female and male at all nests (Spearman rank correlation RS = 0.611, P < 0.001, N = 29), with no significant difference between male and female nest visit rate (Paired-samples T-test T28 = 0.351, P = 0.728). Nest visit frequency was low and quite stable over the day with an average of 0.77 (± 0.10 s.d.) individual visits per hour. The nest visit rate was related to the initial brood size (GLM F1,28 = 5.90, P = 0.02; Table 1) and potentially to the hatching success of a clutch (GLM F1,28 = 3.76, P = 0.06; Table 1). However, nest visit rate was not related to the condition of a brood, the variance in condition across the brood, the breeding density or the initiation date of the breeding attempt (Table 1).

Table 1 Overall nest visit rate and nest visit synchrony in relation to breeding success and environmental factors GLM results with overall nest visit rate as response variable in model 1 and nest visit synchrony as response variable in model 2, least significant factors were removed stepwise, displayed here are the values before removing them from the model.

	1. Overall nest visit rate	2. Nest visit synchrony	
Brood size	F1,26 = 5.90; P = 0.02	F1,27 = 7.56; P = 0.01	
Number fledging	F1,25 = 0.56; P = 0.46	F1,24 = 0.09; P = 0.77	
Hatch success	F1,26 = 3.76; P = 0.06	F1,26 = 2.52; P = 0.12	
%Difference nestling mass	F1,19 = 0.47; P = 0.50	F1,18 = 0.04; P = 0.85	
%Difference nestling condition	F1,20 = 0.00; P = 0.97	F1,20 = 0.51; P = 0.48	
Nesting density	F1,18 = 0.01; P = 0.91	F1,25 = 0.39; P = 0.54	
Initiation date	F1,24 = 0.37; P = 0.55	F1,19 = 0.09; P = 0.77	

On average, partners visited the nest together during 73.3% (± 20.1 S.D.) of visits, which is significantly more than expected by chance (observed proportion of synchronous nest visits vs. the time together at the nest when assuming random nest visit behaviour; paired T-test T = −19.071, P < 0.001, N = 29 (i.e., the pair were both found together at the nest more frequently than by chance, calculated using the actual number of minutes that each individual spent at the nest on each day).

Overall, parents spent an average of 3m24s (± 8m36s s.d.) in the nest cavity during an individual visit (visits scored n = 435). However there were 28 excessively long visits (6% of 435 visits were over ten minutes in duration), with all other visits being substantially shorter in duration. Seventeen of these long visits were by females and 11 by males, of which five were by one particular male and were spread throughout the day. It was not clear what the parent was doing in the nest during these extended visits as the nestlings were over ten days old and the ambient temperature ranged between 20.6 and 34.9°C, suggesting that they are unlikely to have been brooded at this time. When removing these exceptional 28 visits (out of 435), the mean time spent inside the nest per visit was 1m46s ± 50s per visit for males and 1m41s ± 56s (S.D.) for females. These data allow us to calculate the probability of parents being at the nest in the same time, by chance. On average, each parent was present at the nest for approximately 0.067 of the day: 1.46 [average minutes in nest] + 5 [minutes spent waiting in tree before or after feeding]) × 7.72 [average number of nest visits per day]/594 [average minutes filmed per day]. If the parents were working randomly with respect to one another, the likelihood of them both being at the nest at the same time is therefore 0.0045 (0.067 × 0.067).

Although not quantified, anecdotal observations (during our other nest monitoring work) suggested that parents usually arrived and perched in the nest tree together before the first bird entered the nest. The second bird usually waited outside the nest while the first one went inside, and then when the first bird exited the nest it remained in the nest tree and waited while the second bird was in the nest. When the second bird exited the nest, they both usually flew off together.

The proportion of synchronized nest visits by a pair was independent of overall nest visit rate (Spearman rank correlation RS = 0.204, P = 0.288, N = 29). Five pairs always visited the nest together (17% of pairs), with, an average of 7.8 synchronous visits per pair across the day. In the other 24 pairs the average proportion of synchronous visits was 66% with the minimum number of synchronous visits by any pair being 1 out of a total of 11 visits to the nest by both parents.

The proportion of synchronized visits by a pair was positively related to the initial brood size at hatching (GLM F1,28 = 7.56, P = 0.01; Fig. 1), but was not related to other estimates of reproductive success: the proportion of eggs that hatched, number of fledglings, variation across the brood in condition (Table 1).

Figure 1 The correlation between brood size (at hatching) and nest visit synchrony in 29 pairs of long-tailed finch parents while feeding their nestlings (Pearson correlation = 0.47, N = 29, P = 0.011).

Determinants of nest visit synchrony

We were unable to identify any determinants of nest visit synchrony with respect to the characteristics of the breeding attempt, pair or the individual males and females. The proportion of synchronous visits was not affected by nesting density or the date on which the reproductive attempt was initiated (Table 1). Nest visit synchrony was not affected by male (F3,28 = 0.253, P = 0.859) or female age (F2,28 = 2.049, P = 0.149). Visit synchrony was not affected by pair bond duration, i.e., whether a pair had bred together before or not (F11,16 = 1.417, P = 0.396), the number of previous seasons in which the pair had bred before (F15,28 = 1.462, P = 0.249) or the number of times the pair had bred together in the current season (F4,28 = 1.573, P = 0.214).

A small number of pairs were followed during two (N = 6), or three nesting attempts (N = 2). There was no increase in synchrony between the first and the second recorded breeding attempt (Paired T-test T5 = −0.463, P = 0.663; Fig. 2). Three of these pairs had bred together before and did not differ in synchrony between the first and second recorded brood (Paired T-test T2 = 0.144, P = 0.899), the other three pairs had never been recorded as breeding together before but did not show increased synchrony between the first and second brood (Paired T-test T2 = −1.537, P = 0.264). For the two pairs that bred three times together, the synchrony of visits on the third attempt was not different to that during the first (Paired T-test T1 = 0.909, P = 0.530) or second (Paired T-test T1 = 1.320, P = 0.413) attempt.

Figure 2 Nest visit synchrony in multiple attempts during a season for six pairs.

Black lines indicate those pairs that had never bred together before (pair 1–3), grey lines indicate those pairs that had bred together before (pair 4–6).

Discussion

In this study of the long-tailed finch we found that nest visits are very infrequent – each partner visited the nest on average less than once an hour – but the individual visit rate was correlated between partners. Furthermore, the male and female typically arrived together (73% of occasions) and entered the nest individually but one after another. Both the frequency of nest visits and the proportion of synchronised visits are similar to those reported in a couple of other Australian species. In the crimson rosella Platycercus elegans that feeds its offspring seeds, buds, and fruits, parents visited the nest 0.75 times per hour with 63% of visits being synchronous (Krebs, Cunningham & Donnelly, 1999). In the zebra finch, parents visited the nest 0.96 times per hour with 78% of visits being synchronised (Mariette & Griffith, 2012). As in other birds that provision offspring with regurgitated food from the crop, nest visit rate in this species, is likely to have been a poor measure of how much food the parents delivered to the nest, because parents can deliver variable amounts of food in a single visit (Krebs, Cunningham & Donnelly, 1999; Gilby et al., 2011). As a result we will devote most of our discussion to the temporally coordinated behaviour of the male and female.

Variation in the degree of nest visit synchrony by pairs was positively related to the number of chicks that hatched, but was unrelated to all other measures of reproductive success and offspring quality. This might suggest that the degree of synchrony is not primarily about tuning the dynamics of parental care in itself, contrary to what has been suggested. For example, Forbes (1993) hypothesised that by visiting together and ‘clumping’ the distribution of food, parents will be better able to distribute food more equitably amongst their offspring. When food is delivered in a clumped fashion, the most competitive offspring will be overwhelmed and quickly sated by the amount of food available in a short time, ensuring that less competitive offspring are also able to receive some (Forbes, 1993). It has also been suggested that by visiting together, parents could increase the information they have about each other, and the needs of the offspring at the end of the provisioning bout, improving the quality of parental care (Johnstone & Hinde, 2006). Further work that could be usefully conducted in this and other species with synchronised visits to the nest would be to examine the extent to which parents practise strategic feeding positions or the distribution of food to the nestlings when they feed nestlings alone or in concert with one another (Lessells et al., 2006; Smiseth et al., 2003; Dickens & Hartley, 2007).

However, although these ideas are certainly worthy of further investigation in this and other species we could find no support for them in these observations. Neither the overall quality of broods or the variance in offspring quality across a brood were related to the level of coordination by parents. Furthermore, we were unable to identify any predictors of the degree of nest visit synchrony with respect to a number of ecological and pair characters such as how long a pair had apparently been together suggesting that its variation was not driven by familiarity or experience of the partnership as might have been expected (Fowler, 1995; Black, 1996). Whilst coordination of behaviour might be one of the benefits of forming long-term partnerships, it is possible that this can be achieved relatively quickly and may be achieved through the pairing of compatible personalities (Schuett, Dall & Royle, 2011), or through private acoustic duets as seen in the zebra finch (Elie et al., 2010).

Long-tailed finch nests are vulnerable to predation and we have directly witnessed predation of nests by Olive Pythons Liasis olivaceus and Pied Butcherbirds Cracticus nigrogularis (van Rooij & Griffith, 2011). Many other nesting attempts (>60%) in the study area fail to produce any fledglings, but it is difficult to determine the extent to which this is due to predation of nest contents or the result of adult desertion (van Rooij & Griffith, 2011). Nonetheless, it is likely that nest predators account for nests other than the ones that we encountered during the act of predation. As well as predating the chicks these predators are also capable of predating an adult bird that is caught in the cavity nest (which typically only has a single entrance). Synchronised nest visits by parents provide two important routes to reducing predation of the nestlings and the adults themselves. By synchronizing visits, adults will minimise the number of occasions throughout the day when there is activity and loud chick begging calls around the nest (Raihani et al., 2010). A pair that deliver all of their food together will halve the number of feeding bouts throughout the day compared to a pair that visit the nest separately. Second, when the two parents visit the nest together (and enter one at a time) then they can effectively look out for one another, with a ‘sentinel’ outside alerting its partner inside the nest to any coming predators. In the zebra finch, Mainwaring & Griffith (2013) identified this sentineling behavior for the first time in a socially monogamous species and experimentally demonstrated how effective it was at enabling the bird inside the nest to receive advance warning and leave the nest cavity before a predator arrived. Nest visit synchronization is therefore likely to reduce predation of both offspring and adults around the nest. This is a potentially important component that may only affect a relatively small number of nests each year (and hence not get easily detected by a study such as this), but have catastrophic consequences for the reproductive event and the future success of the adults. Therefore, this mode of selection may be a strong force acting on coordinated patterns of parental care, even though it would take a larger sample to detect an affect of nest visit synchrony on actual predation rates.

It is important also to consider that nest visit synchrony might be primarily driven by coordinated and synchronized activity away from the nest. In the zebra finch the pair associate very closely with one another throughout the day away from the nest and indeed even during periods when they are not breeding (Zann, 1996; Mariette & Griffith, 2012). Therefore nest visit synchrony could be a side effect of the pair feeding together. Synchronization might be easier for granivorous species because the nature of the food source enables a pair to forage efficiently next to one another, compared to predatory species that have to hunt very mobile prey which are more efficiently gathered by individuals hunting alone. We are unable to investigate it here as we have no data on foraging behaviour, however it is possible that foraging patterns relating to local habitat might explain some of the variation in synchronous nests visits that we see across the pairs in this study.

In summary, our study has revealed a pattern of highly coordinated behavior by a breeding pair that results in a high level of synchronized provisioning visits to the nest. It is likely that this behavior reduces vulnerability of adults and offspring to predation but it may emerge from a high level of synchrony during foraging away from the nest and throughout the year. Either way, our study highlights the importance of considering the cooperative elements of behavior in socially monogamous species, something that has been neglected over the past few decades in favour of more fashionable ideas about sexual conflict.

Supplemental Information

Supplemental Information 1 Long-tailed finch and nest

Adult long-tailed finch emerging from nest.

Click here for additional data file.

We would like to thank James Brazill-Boast, Sarah Pryke and all volunteers for invaluable assistance in the field.

Additional Information and Declarations

Competing Interests

Author Contributions

Animal Ethics

Field Study Permissions

The authors declare that there are no competing interests.

Erica P. van Rooij performed the experiments, analyzed the data, contributed reagents/materials/analysis tools, wrote the paper.

Simon C. Griffith conceived and designed the experiments, analyzed the data, contributed reagents/materials/analysis tools, wrote the paper.

The following information was supplied relating to ethical approvals (i.e., approving body and any reference numbers):

This study was approved by the Macquarie University Animal Ethics Committee (Approval number 2007/038).

The following information was supplied relating to ethical approvals (i.e., approving body and any reference numbers):

The Western Australia Department of Environment and Conservation (no. BB 002563).

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
