# Peer review of "Synchronised provisioning at the nest: parental coordination over care in a socially monogamous species"

_PeerJ, doi:10.7717/peerj.232_

## Round 0.1 · original submission · Minor Revisions

We have now received two reviews on your study, both of which were very positive and suggested minor, but highly relevant, revisions. Your manuscript will be thus become acceptable for publication in PeerJ once you’ve addressed these comments and suggestions.

·

Basic reporting

The article satisfies your standards in all respects. The introduction and background are particularly helpful.

Experimental design

No comments.

Validity of the findings

The inferences, although somewhat speculative, are very reasonable and nicely argued.

Additional comments

This is a very nice study and the manuscript is clear and easy to follow in all of its sections. The analyses are appropriate, the inferences are persuasively presented and, even though there are no data on predation, I found the conclusions persuasive. Coordination in nest visits may well be an important aspect of biparental care in some birds and this study encourages us all to consider this positive side of biparental relations which, as the authors point out, has been neglected up till now. Variation in synchrony among pairs is an issue raised by the authors but not explained by any of the analyses, so the reader is left wondering. Maybe it could be explained, speculatively, by variation in their foraging patterns, itself due to variation in the territories in which they forage.

Abstract. The last sentence does not do justice to what is said in the Discussion and probably should mention sentinel behavior.
95. “bias” rather than “potential bias”, surely.
110. “condition” of who?
113-114 Seems overly tautological.
136. If hatching is asychronous then you need to explain how you identified individual chicks, which is important for the credibility of measures of condition etc. Chicks weren’t banded until age 10 d, so how did you distinguish individuals at that age?
144. What does “same area” refer to?
146-152. The type of monitoring that these data are based on is not described and should be described briefly
157. What is E.P.vR.?
171. “scored” rather than “recorded”
182-183. Were these probabilities of expected coincident visits also based on a criterion of <5min equals coincidence, as seems to be required?
199-200. Too concise for me; I don´t get it.
202. Sample size would be helpful here.
204. Maybe “substantially” rather than “significantly”
267. For “as” maybe substitute “contrary to what has”, to avoid ambiguity.
277. What does “asynchrony in offspring quality” refer to? Need to rephrase.
302. “effect” should be “affect”
303 “having” should be “have”.

Reviewer 2 ·

Basic reporting

This is a well-written manuscript in which the authors present data on a poorly-known Australian passerine: the long-tailed finch (Poephila acuticauda).The authors report an insteresting behaviour (synchronised provisioning) and their possible causes. Overall, I enjoyed the manuscript and I have no major concerns about the approach or interpretations. I think this study is a valuable contribution that deserves to be published. My minor comments are below:

L34 A more recent rewiew maybe quoted here (Royle, Smiseth & Kölliker. 2012. The Evolution of Parental Care. Oxford University Press)

L118-120 This information is provided in “Ethics”. Remove from “Methods”

L121 wooden nestboxes rather than “boxes”

L138-139 Why you do not use intra-brood coefficient of variation of nestling body mass (see e.g. Orell 1993)?

L139 In some species, hatching asynchrony leads to the presence of “runts” (small nestlings with a very low survival rate -I never ring these chicks-). What is about it?

L143 I think “Breeding density” should be “local breeding density”

L144 “[…] as the number of active nests in the same area as the focal nest” Non clear. What is the same area? Please provide a more accurate description. For instance, the number of active nests in a 100 m radius (≤ 100 m).

L158-159 You should clarify at which age parents were captured (e.g. adults were captured on day 7-8 post-hatching…)

L197 nesting or breeding density?

L208 Celsius instead of Celcius

L228 …was POSITIVELY related to…

L278 Adults were captured, aged and marked but it seems that they were not measured (tarsus length, body mass). I think that adult characteristics (e.g. body condition) could be included as potential predictors of the degree of nest visit synchrony (for instance, females in poor condition could be be escorted more often by their partners)

L286-287 Please, add predation rate for the study area

Fig. 1. The figure legend is a bit dull, please add more information (species, n, etc.)

Concluding remark. (suggestion on further work) It would be interesting to explore parental feeding positions (see Lessells et al. 2006) and food distribution among siblings (Lessells 2002, Parker et al. 2002, Smiseth et al. 2003, Dickens & Hartley 2007) when they work in concert (synchronised provisioning) or alone. By this way, authors may determine if parents employ the same food allocation rules when they feed their young alone or accompanied

Experimental design

No comments

Validity of the findings

No comments

---

## Round 0.2 · accepted · Accept

You have done a great job in answering and integrating the comments previously provided on your manuscript. It is now acceptable for publication in Peer J.